# Preimplantation Genetic Diagnosis in Hereditary Hearing Impairment

**DOI:** 10.3390/diagnostics11122395

**Published:** 2021-12-20

**Authors:** Hsin-Lin Chen, Pei-Hsuan Lin, Yu-Ting Chiang, Wen-Jie Huang, Chi-Fang Lin, Gwo-Chin Ma, Shun-Ping Chang, Jun-Yang Fan, Shin-Yu Lin, Chen-Chi Wu, Ming Chen

**Affiliations:** 1Department of Otolaryngology, National Taiwan University Hospital, Cancer Center Branch, Taipei 10672, Taiwan; nilnishchen@gmail.com; 2Department of Otolaryngology, National Taiwan University Hospital Yunlin Branch, Yunlin 63247, Taiwan; ru3au3@gmail.com; 3Department of Otolaryngology, National Taiwan University Hospital, Taipei 10002, Taiwan; ashley.chiang413@gmail.com; 4Graduate Institute of Medical Genomics and Proteomics, National Taiwan University College of Medicine, Taipei 10051, Taiwan; 5Department of Otorhinolaryngology, Head and Neck Surgery, Changhua Christian Hospital, Changhua 50046, Taiwan; d0700@cch.org.tw (W.-J.H.); 84850@cch.org.tw (J.-Y.F.); 6Department of Genomic Medicine and Center for Medical Genetics, Changhua Christian Hospital, Changhua 50046, Taiwan; enokialin@gmail.com (C.-F.L.); 128729@cch.org.tw (G.-C.M.); 71914@cch.org.tw (S.-P.C.); 7Department of Obstetrics and Gynecology, National Taiwan University Hospital, Taipei 10041, Taiwan; 8Department of Medical Research, National Taiwan University Hospital Hsin-Chu Branch, Hsinchu 30261, Taiwan; 9Department of Medical Science, National Tsing Hua University, Hsinchu 30013, Taiwan; 10Department of Biomedical Science, Dayeh University, Changhua 51591, Taiwan; 11Medical College Provisional Office, National Chung Hsing University, Taichung 40227, Taiwan

**Keywords:** preimplantation genetic diagnosis (PGD), hereditary hearing impairment, deafness-associated genes, *SLC26A4*, *GJB2*, *OTOF*

## Abstract

Sensorineural hearing impairment is a common sensory deficit in children and more than 50% of these cases are caused by genetic etiologies, that is, hereditary hearing impairment (HHI). Recent advances in genomic medicine have revolutionized the diagnostics of, and counseling for, HHI, including preimplantation genetic diagnosis (PGD), thus providing parents-to-be with better reproductive choices. Over the past decade, we have performed PGD using the amplification refractory mutation system quantitative polymerase chain reaction (ARMS-qPCR) technique in 11 couples with a history of HHI, namely eight with *GJB2* variants, one with *OTOF* variants, one with *SLC26A4* variants, and one with an *MITF* variant. We demonstrated that PGD can be successfully applied to HHI of different inheritance modes, namely autosomal dominant or recessive, and phenotypes, namely syndromic or non-syndromic HHI. However, certain ethical concerns warrant scrutiny before PGD can be widely applied to at-risk couples with a history of HHI.

## 1. Introduction

Sensorineural hearing impairment (SNHI) is the most common sensory deficit in children. Approximately 0.2% of newborns have permanent bilateral SNHI, and its prevalence increases during childhood and adolescence [1,2]. Childhood SNHI can be divided into hereditary hearing impairment (HHI) or acquired hearing impairment, including congenital infections, prematurity, kernicterus, and perinatal insults [2,3,4]. It is estimated that approximately two-thirds of SNHI children in developed countries have HHI [5]. To date, more than 200 HHI-associated genes have been identified (http://hereditaryhearingloss.org, accessed on 8 November 2021).

In many hereditary diseases, determining the genotypes in embryos by using preimplantation genetic diagnosis (PGD) techniques has been demonstrated to be an effective procedure for preventing the transmission of genetic abnormalities across generations [6,7]. However, there are still only limited studies regarding the use of PGD to address HHI. Consequently, we present this report of our experience over the past decade with the use of PGD in families with HHI at two tertiary teaching hospitals and discuss the associated ethical issues.

## 2. Materials and Methods

### 2.1. Subjects

From 2009–2021, a total of 11 couples requested PGD for HHI at Changhua Christian Hospital (*n* = 9, Case 1–9) and National Taiwan University Hospital (*n* = 2, Case 10–11). An experienced geneticist provided comprehensive counseling to the couples and informed them of the possibility that several treatment cycles could be needed before pregnancy was achieved. The odds of misdiagnosis inherent to single-cell PCR were discussed, and subsequent genotype confirmation of the pregnancy using conventional prenatal diagnosis was recommended.

### 2.2. PGD and Prenatal Confirmation Process

To fulfill the need for timely and overnight diagnosis of fresh embryo transfer, we utilized the amplification refractory mutation system quantitative polymerase chain reaction (ARMS-qPCR) genotyping strategy and conformed to the international standard for using amplification-based methodology for PGD [8,9]. The detailed ARMS-qPCR procedure is based on our previously published literature [10] and is applied to PGD of autosomal recessive aromatic L-amino acid decarboxylase deficiency (AADC). In brief, two primer sets were used, firstly for amplification by a duplex-nested PCR with the consistent PCR conditions, and secondly, two sequence-specific forward primers for ARMS-qPCR were modified with a mismatch at the penultimate nucleotide position of the mutation site to increase the specificity of the PCR reaction. All primers used are listed in Appendix A Table A1. Finally, wild-type and mutant alleles were distinguished by assessing the crossing points (Cp) value through the qPCR performed on a Roche LC 480 system (Basel, Switzerland). A trophectoderm biopsy was performed at the 5/6-day blastocyst stage. If confirmation was received that the embryos were inappropriate for transfer, they were discarded. 

All pregnant women underwent confirmatory invasive prenatal diagnosis to avoid the live birth of babies affected by HHI. Diagnostic informed consent documents were obtained from all couples who were subjected to the clinical preimplantation genetic studies. As this study was a retrospective chart review, the request for the waiver of additional informed consent was approved by the research ethics committees of both hospitals.

## 3. Results

### 3.1. Overall Outcomes

The genotypes and PGD processes of the 11 couples are shown in Table 1. Of the 11 couples, Case 1–10 had normal hearing but already had offspring that suffered from HHI. The causative variants in deafness genes were confirmed, namely eight with bi-allelic *GJB2* variants, one with bi-allelic *SLC26A4* variants, and one with bi-allelic *OTOF* variants. In addition, a pathogenic *ACADVL* variant was incidentally identified in Case 10. The last couple included in the study (Case 11) were both hearing-impaired. The woman had non-syndromic profound SNHI caused by bi-allelic *MYO15A* variants and the man had Waardenburg syndrome caused by a dominant *MITF* variant. Therefore, the indications for PGD in these 11 cases included *GJB2* variants (*n* = 8), *SLC26A* variants (*n* = 1), *OTOF* variants (*n* = 1), and *MITF* variants (*n* = 1).

These 11 couples underwent a total of 136 oocyte retrievals for in vitro fertilization (IVF) with PGD (Table 1). A total of 60 embryos were sent for PGD and the successful diagnosis rate was 100% (60/60). 

All the embryos classified as affected were confirmed before being discarded; therefore, the false positive rate was 0%. Four patients delivered normal, healthy babies and the live birth rate (take-home baby rate) was 2.94% (4/136) per oocyte retrieval and 30.76% (4/13) per transfer cycle (Table 1). All the babies that were born were confirmed to be unaffected by hearing loss and heritable variants through postnatal genotyping and newborn hearing screening.

### 3.2. An Example with GJB2 and ACADVL Variants

Due to their family history of hearing impairment, the couple in Case 9 (Figure 1) underwent gene testing. The couple consisted of a 39-year-old woman with the heterozygous *GJB2* c.427C>T variant and a man who was heterozygous for the *GJB2* c.109G>A variant, as well as heterozygous for *ACADVL* c.277+2T>G as an incidental finding (Table 1). The *ACADVL* gene encodes very long-chain acyl-CoA dehydrogenase (VLCAD), and recessive *ACADVL* variants result in VLCAD deficiency. VLCAD deficiency is a fatty acid oxidation defect that prevents the body from converting certain fats to energy, particularly during fasting, and may cause sudden unexpected deaths in infants. Signs and symptoms of VLCAD deficiency typically appear during infancy or early childhood and can include hypoglycemia, lethargy, muscle weakness, liver abnormalities, and life-threatening heart problems [11]. The *ACADVL* c.277+2T>G variant was classified as “likely pathogenic” according to the American College of Medical Genetics and Genomics (ACMG) guidelines. During their second pregnancy, they underwent prenatal screening with whole exome sequencing, which identified compound heterozygous *GJB2*:c.[109G>A];[427C>T] in the fetus. During PGD, three embryos with good morphology were selected and biopsied from blastocysts on day 6. Trophectoderm cells were examined by ARMS-qPCR to detect the presence of the *GJB2* c.109G>A and c.427C>T variants and presence was confirmed by Sanger sequencing. One embryo was selected for transfer because it did not have these variants. The couple underwent successful IVF, resulting in pregnancy.

## 4. Discussion

In this study, we reported our experience with PGD in 11 couples with a family history of HHI. Of these 11 couples, eight had *GJB2* variants (DFNB1), one had *OTOF* variants (DFNB9) that were all linked to autosomal recessive non-syndromic HHI, one had *SLC26A4* variants that were linked to both autosomal recessive non-syndromic HHI (DFNB4) and autosomal recessive Pendred syndrome, and one had an *MITF* variant that was linked to autosomal dominant Waardenburg syndrome. Our results demonstrated that PGD can be applied to HHI of different inheritance modes (autosomal dominant or autosomal recessive) and phenotypes (syndromic or non-syndromic HHI).

Prior to our study, there were only four reports regarding the use of PGD in families with HHI (Table 2) [12,13,14,15]. Altarescu et al. developed a single-cell multiplex PGD protocol to analyze blastomeres and polar bodies and claimed that blastomere analysis was preferable to polar body analysis for *GJB2* and *GJB6* variants [12]. Wu et al. conducted single-cell testing using GenomiPhi technology and primer extension mini-sequencing for the *SLC26A4* c.919-2A>G variant [13]. Yazdi et al. applied single-cell Sanger sequencing to identify *GJB2* variants [14]. Hao et al. performed targeted next-generation sequencing (NGS)-based PGD using an ion torrent platform [15]. In this study, we adopted the ARMS-qPCR technique, a cost-efficient and time-saving method for PGD [16]. To the best of our knowledge, this study represents the largest and most diverse series in the literature on the application of PGD to HHI.

However, the ethical concerns related to the application of PGD to address HHI merit discussion. Firstly, the debate on using PGD for hearing impairment centers around the different interpretations of “disability” by stakeholders. The definitions of “disability” seem to imply different normative judgments about parental reproductive choices [17,18]. The development of various hearing-assisting devices, such as hearing aids and cochlear implants, has substantially mitigated the impact of SNHI on affected children [2]. Even for children with profound SNHI, cochlear implantation is still effective and can help them participate in mainstream educational and occupational systems [19]. Secondly, the phenotypic heterogeneity and the lack of clear genotype–phenotype correlations in HHI make it difficult to perform counseling prior to PGD. For instance, it has been demonstrated that patients with the same *GJB2* genotype may exhibit wide variations in severity of hearing impairment [20,21,22]. In particular, some patients with bi-allelic *GJB2* variants, which were classified as pathogenic according to the ACMG guidelines (for example p.V37I and p.M34T), might have normal hearing or very mild SNHI throughout their lifetime [23,24]. Offering reproductive advice for such genotypes, which result in normal or mild phenotypes, may introduce ethical dilemmas. 

Meanwhile, the increasing popularity of NGS in the diagnosis of HHI may pose additional difficulties in the application of PGD. Recent advances in NGS technologies, including targeted screening panels, whole exome sequencing, and whole genome sequencing, have revolutionized the clinical management of HHI in terms of genetic diagnosis, genetic counseling, and response to diagnosis [25,26]. Although there is strong evidence that the use of NGS can significantly increase the diagnostic yield of genetic examination for HHI [27], it also leads to the identification of a large number of variants of uncertain significance (VUS) [28,29]. The associations between these VUS and the phenotypes of SNHI remain largely unclear and may contribute to over-anxiety in some parents and the potential abuse of PGD [30]. 

When PGD was first implemented, the procedure was considered to be extremely complex and associated with a high risk of errors. Currently, rapid advances in assisted reproductive technology have been made, and the efficacy and accuracy of PGD have gradually increased. Current methods of PGD include fluorescent in situ hybridization, comparative genomic hybridization, single nucleotide polymorphism analysis, and NGS. Direct diagnostic methods include PCR and whole genome amplification [31,32]. In this study, the majority of the cases were diagnosed using the ARMS-qPCR method to provide a rapid and accurate diagnosis. In the future, if the turnaround time of NGS is shortened further and the interpretation of VUS becomes more precise, NGS-based PGD may play a more important role in clinical practice.

## 5. Conclusions

We reported our experience with PGD in 11 couples with a family history of HHI, namely eight with *GJB2* variants, one with *OTOF* variants, one with *SLC26A4* variants, and one with an *MITF* variant. Our results demonstrated that PGD can be potentially applied to HHI of different inheritance modes, namely autosomal dominant or autosomal recessive, and phenotypes, namely syndromic or non-syndromic HHI.

## Figures and Tables

**Figure 1 diagnostics-11-02395-f001:**
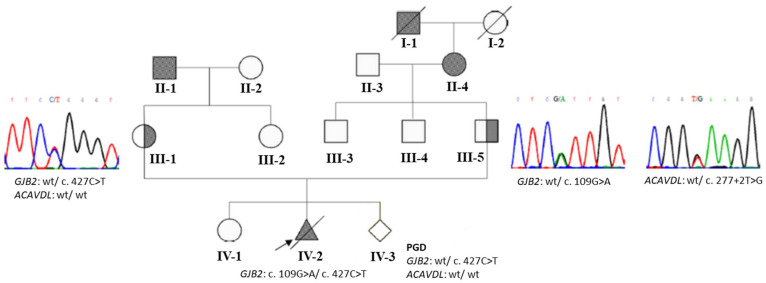
Pedigree and genotyping of an index family with *GJB2* and *ACADVL* variants. The woman of the Case 9 couple (III-1) carried a heterozygous *GJB2* c.427C>T variant, whereas the man (III-5) was heterozygous for the *GJB2* c.109G>A variant, as well as heterozygous for *ACADVL* c.277+2T>G as an incidental finding. During their second pregnancy, they underwent whole exome sequencing for prenatal screening, which identified compound heterozygous *GJB2*:c.[109G>A];[427C>T]. They used PGD and pregnancy was achieved after transferring one unaffected embryo.

**Table 1 diagnostics-11-02395-t001:** PGD of HHI in our study.

Case	Maternal Age	Maternal Disease Status	Maternal Genotype	Paternal Disease Status	Paternal Genotype	Disease Gene	Inheritance Mode	Mutation Type	Diagnostic Methodology	Oocyte Retrievals	Embryos Diagnosed	Diagnostic Results (Unaffected/ Affected)	Embryos Transferred	Pregnancy Outcome
1	34	C	c.2168A>G (p.H723R)	C	c.1229C>T (p.T410M)	*SLC26A4*	AR	S	ARMS qPCR	6	6	3/3	3	failed
2	30	C	c.109G>A (p.V37I)	C	c.235delC (p.L79fs)	*GJB2*	AR	S + D	ARMS qPCR	13	2	1/1	1	failed
3	32	C	c.109G>A (p.V37I)	C	c.109G>A (p.V37I)	*GJB2*	AR	S	ARMS qPCR	7	7	1/6	1	failed
4	31	C	c.109G>A (p.V37I)	C	c.235delC (p.L79fs)	*GJB2*	AR	S + D	ARMS qPCR	9	5	5/0	2	1 boy
5	33	C	c.235delC (p.L79fs)	C	c.109G>A (p.V37I)	*GJB2*	AR	S + D	ARMS qPCR	11	5	2/3	1	1 boy
6	33	C	c.109G>A (p.V37I)	A	homozygous c.109G>A (p.V37I)	*GJB2*	AR	S	ARMS qPCR	28	11	7/4	1	failed
7	35	C	c.5098G>C (p.E1700Q)	C	c.5197G>A (p.E1733K)	*OTOF*	AR	S	ARMS qPCR	20	3	3/0	1	failed
8	33	C	c.109G>A (p.V37I)	C	c.235delC (p.L79fs)	*GJB2*	AR	S + D	ARMS qPCR	11	8	6/2	2	1 girl
9	39	C	GJB2 c.427C>T (p.R143W)	C	GJB2 c.109G>A (p.V37I) heterozygous ACADVL c.277+2T>G	*GJB2/ ACADVL*	AR	S	ARMS qPCR	14	3	1/2	1	1 boy
10	39	C	c.235delC	C	c.299_300delAT(p.His100fs)	*GJB2*	AR	D	STR analysis	2	1	1/0	NI	NI
11	31	A	*MYO15A* c.3524dup; 6956+1G>A/*GJB2* c.299_300delAT carrier	A	*MITF* c.1052C>T	*MITF*	AD	S + D	STR analysis	15	9	6/3	NI	NI

AD, autosomal dominant; ARMS-qPCR, amplification refractory mutation system quantitative polymerase chain reaction; AR, autosomal recessive; C, carrier; A, affected; S, substitution; D, deletion; NI, no implantation or pregnancy outcome; STR, Sanger sequencing short tandem repeat.

**Table 2 diagnostics-11-02395-t002:** Previous reports applying PGD to hereditary hearing impairment.

Year	Disease Genes	PGD Method	Strategy	Success Case No.	Place	Reference
2009	*GJB2/GJB6*	Polar body and blastomere PCR	Direct diagnosis	8	Israel	[12]
2010	*SLC26A4*	Single-cell mini-sequencing	Direct diagnosis	1	Taiwan	[13]
2018	*GJB2*	Single-cell Sanger sequencing	Direct diagnosis	1	Iran	[14]
2018	*SLC26A4*	Next-generation sequencing	Direct diagnosis	1	China	[15]

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
