# Peer review of "Preimplantation Genetic Diagnosis in Hereditary Hearing Impairment"

_diagnostics, 2021, doi:10.3390/diagnostics11122395_

Round 1
Reviewer 1 Report
Table 1, line 9, name of gene ACADVL must be write without hyphenation.
lines 121-124 of text must be justified
line 121 ” morphologies” must be replaced with ”morphology”
I consider that table 2 is not requested because the information is presented in detail in text (lines 142-152).
the sentence: „Offering reproductive advice for such genotypes may introduce ethical dilemmas.” - lines 168-169 - must be reformulated because it is not very clear.
Author Response
Thanks for your valuable critics and suggestions to significantly improve the previous version of our article. We have tried to revise our manuscript accordingly.
Point 1: Table 1, line 9, name of gene ACADVL must be written without hyphenation.
Response 1: We change to the correct form.
Point 2: lines 121-124 of text must be justified
Response 2: The text is justified.
Point 3: line 121 ” morphologies” must be replaced with ”morphology”
Response 3: We replace the word with “morphology.”
Point 4: I consider that table 2 is not requested because the information is presented in detail in the text (lines 142-152).
Response 4: Thank you for your suggestion. Although the information is presented in detail in the text, we decide to keep table 2 for better reading confluency.
Point 5: the sentence: „Offering reproductive advice for such genotypes may introduce ethical dilemmas.” - lines 168-169 - must be reformulated because it is not very clear.
Response 5: We agree, we reformulate the text to “offering reproductive advice for such genotypes which result in normal or mild phenotypes may introduce ethical dilemmas.”
Reviewer 2 Report
- Figure 1 legend - too detailed; the same story from text is repeated; the international symbols for pedigree should be known by any physician, I don't think they should be explained;
- Table 1 looks too crowded; I think the term patient should be replaced with affected; just to have less text, carrier could be replaced with C and affected replaced with A and the abbreviations could be defined in the lower part; the word heterozygous is very long - if the individual is a carrier, it is understood that he/she is heterozygous;
- Reference 13 doesn't have a doi number;
- Reference 10 is not cited
Author Response
Thanks for your valuable critics and suggestions to significantly improve the previous version of our article. We have tried to revise our manuscript accordingly.
Point 1: Figure 1 legend - too detailed; the same story from text is repeated; the international symbols for pedigree should be known by any physician, I don't think they should be explained.
Response 1: We agree. We delete the explanation for international symbols for pedigree.
Point 2: Table 1 looks too crowded; I think the term patient should be replaced with affected; just to have less text, carrier could be replaced with C and affected replaced with A and the abbreviations could be defined in the lower part; the word heterozygous is very long - if the individual is a carrier, it is understood that he/she is heterozygous.
Response 2: Thank you for your suggestion. We try to improve the layout of the table. Thus, we delete heterozygous if the individual is a carrier.
Point 3: Reference 13 doesn't have a doi number
Response 3: We add doi:10.14715/CMB/2018.64.9.11.
Point 4: Reference 10 is not cited.
Response 4: Reference is cited in line 124, “and life-threatening heart problems [10].”
Reviewer 3 Report
In this manuscript, the authors report their experience with PGD in eleven families with Hearing loss. Recessive and dominant inherited models were analyzed. They postulate that the PGD could be applied to hearing loss.
Major considerations:
Materials and methods need to be improved. You should include more information, for example, primers used and PCR conditions…
I wonder how it is possible to detect a mutation in ACADVL, I suppose you have used specific primers for the gene you analyze. Could you clary it, please?
Minor considerations
In table 1, mutation type not all the mutation analyzed are substitution, Some of them are deletion, for example, c.235del C, Please check it.
Author Response
Thanks for your valuable critics and suggestions to significantly improve the previous version of our article. We have tried to revise our manuscript accordingly.
Point 1: Materials and methods need to be improved. You should include more information, for example, primers used and PCR conditions.
Response 1: We agree. We include more information if the material and methods. All primers used are listed in Appendix A.
Point 2: I wonder how it is possible to detect a mutation in ACADVL, I suppose you have used specific primers for the gene you analyze. Could you clary it, please?
Response 2: The primers used to detect ACADVL are shown in Appendix A.
Point 3: In table 1, mutation type not all the mutation analyzed are substitution, some of them are deletion, for example, c.235del C, please check it.
Response 3: We check and correct the content of table 1 again.